# Pollution index of waterfowl farm assessment and prediction based on temporal convoluted network

Jiande Huang [1,2,3,4,5,6], Shuangyin Liu[1,2,3,4,5,6]* *, Shahbaz Gul Hassan[1,2,3,4,5,6]* *, Longqin Xu[1,2,3,4,5,6]

**1** College of Information Science and Technology, Zhongkai University of Agriculture and Engineering, Guangzhou, China, **2** Smart Agriculture Engineering Technology Research Center of Guangdong Higher Education Institues, Zhongkai University of Agriculture and Engineering, Guangzhou, China, **3** Guangzhou Key Laboratory of Agricultural Products Quality, Safety Traceability Information Technology, Zhongkai University of Agriculture and Engineering, Guangzhou, China, **4** Guangdong Provincial Agricultural Products Safety Big Data Engineering Technology Research Center, Zhongkai University of Agriculture and Engineering, Guangzhou, China, **5** Guangdong Province Key Laboratory of Waterfowl Healthy Breeding, Guangzhou, China, **6** Academy of Smart Agricultural Engineering Innovations, Zhongkai University of Agriculture and Engineering, Guangzhou, China

☯ These authors contributed equally to this work.
* shuangyinliu@zhku.edu.cn (SL); mhasan387@zhku.edu.cn (SGH)

**Data Availability Statement:** All relevant data are within the paper and its Supporting Information files.

**Funding:** This work was supported in part by the National Natural Science Foundation of China under Grants 61871475, 61471133, and 61571444,

## Abstract

Environmental quality is a major factor that directly impacts waterfowl productivity. Accurate prediction of pollution index (PI) is the key to improving environmental management and pollution control. This study applied a new neural network model called temporal convolutional network and a denoising algorithm called wavelet transform (WT) for predicting future 12-, 24-, and 48-hour PI values at a waterfowl farm in Shanwei, China. The temporal convoluted network (TCN) model performance was compared with that of recurrent architectures with the same capacity, long-short time memory neural network (LSTM), and gated recurrent unit (GRU). Denoised environmental data, including ammonia, temperature, relative humidity, carbon dioxide ($CO_2$), and total suspended particles (TSP), were used to construct the forecasting model. The simulation results showed that the TCN model in general produced a more precise PI prediction and provided the highest prediction accuracy for all phases (MAE = 0.0842, 0.0859, and 0.1115; RMSE = 0.0154, 0.0167, and 0.0273; R2 = 0.9789, 0.9791, and 0.9635). The PI assessment prediction model based on TCN exhibited the best prediction accuracy and general performance compared with other parallel forecasting models and is a suitable and useful tool for predicting PI in waterfowl farms.

## Introduction

In response to the growing global demand for food, the Chinese waterfowl industry has grown to be a leader in both goose meat and goose egg production [1]. To meet this demand,
one mechanism to increase production is to increase housing and manage more geese. However, as the scale of intensive culture increases, there is a growing concern in China that

awarded to Dr. Shuangyin Liu, in part by the special project of laboratory construction of Guangzhou Innovation Platform Construction Plan under Grant 201905010006 awarded to Dr. Shuangyin Liu, Guangzhou key research and development project under Grant 202103000033 awarded to Dr. Shuangyin Liu, Guangdong Science and Technology Plan of Project under Grant 2017B0101260016 awarded to Dr. Shuangyin Liu, National Key Technologies R & D Program of China under Grant 2016YFD0500510 awarded to Dr. Shuangyin Liu, Guangdong key research and development project under Grant 2020B0202080002 awarded to Dr. Shuangyin Liu, the foundation for High-level Talents in Higher Education of Guangdong Province under Grants 2017GCZX0014, 2016KZDXM0013, 2017KTSCX094, and 2018LM2168 awarded to Dr. Shuangyin Liu, and Beijing Natural Science Foundation under Grant 4182023 awarded to Dr. Shuangyin Liu. The funders had no role in study design, data collection and analysis, decision to publish, or preparation of the manuscript.

**Competing interests:** The authors have declared that no competing interests exist.

waterfowl should be raised under conditions that promote animal welfare since productivity is related to environmental conditions in which the waterfowl are raised [1]. For example, the organic matter present in excreta and/or litter resulting in pollutant production, such as $CO_2$, ammonia, and TSP, will not only impair geese and staff health but also have important consequences for society once the pollutants are out in the atmosphere [2–4]. According to this, to maintain and control optimal conditions for survival and good goose growth, establishing a habitat that is closer to a standard ecosystem is justified [4].

Labor shortages and increasing biosecurity practices will make it more difficult for producers to monitor and manage the production, health, and welfare status of all of their birds. Employing modern poultry management technology is necessary to increase production [5]. An example of how modern management technology can be used to monitor and control the poultry house environment is exemplified by humidity regulation via ventilation rate changes mediated by relative humidity sensors, as relative humidity is one of the more important environmental aspects of a poultry house [5]. In addition, more advanced systems are being researched. Bustamante et al. used a multisensor system to effectively assess barn ventilation system function by tracking temperature, air velocity and differential pressure in broiler houses [6]. Hanif et al. proposed an internet of things technology-based protection and monitoring of the environment of a poultry house to monitor the environment-related parameters such as air temperature, air humidity, CO2 level of concentration and ammonia concentration, which has been implemented successfully, leading to a safe environment and profit for the poultry industry [7]. The techniques mentioned above are both solutions for real-time environmental monitoring and control; however, relying on hardware monitoring in real time cannot capture the trend of environmental changes [8], and it is easy to miss the best time for adjustment, which leads to waterfowl health damage and property loss.

Cultivation environment forecasting has been studied for many years and has made some achievements in aquaculture and livestock breeding [9–10]. The technique estimates or predicts the future changes in target variables that cannot be obtained directly. For example, Jackman et al. generated a prediction model by using sensor inputs of relative humidity, $CO_2$, temperature, and ammonia for environmental parameter and crop yield prediction [11]. In waterfowl production, a system such as this would allow for actions to be taken sooner by farmers if the environment is projected to be bad. However, few studies have applied prediction in waterfowl breeding. Therefore, it is necessary to apply environmental prediction technology to waterfowl production to fill this gap.

The PI is a simple and easy assessment method for assessing environmental quality, and precise environmental quality assessment and prediction allow better environmental management practices and contribute to a more sustainable environmental management approach [12–17]. In recent years, many studies on pollution models have been carried out to evaluate the quality of the environment based on the PI method, such as river quality status assessment and prediction [12, 14–16], air quality status assessment and prediction [17–19], and soil pollution assessment and prediction [20, 21]. Forecasting the concentration of water, air, and soil pollutants is an effective method for protecting public health, productivity, and travel by providing early warnings of harmful pollutants. Likewise, it is also important to assess and predict PI in advance for waterfowl farms so that, if necessary, the producer can intervene more quickly by using management practices to ensure a healthy context.

In general, PI formulations include lengthy computations and thus require considerable time and effort. Additionally, waterfowl environment pollutant data are dynamic, complex, and have high temporal and spatial variability [22, 23], and traditional forecasting methods such as multiple linear regression and autoregressive integrated moving average models have

poor performance. Hence, a method to calculate PI in an efficient and precise way is required. Such approaches may benefit farmers when assessing and managing environmental quality.

Over the past few decades, artificial intelligence has been increasingly applied to solve various environmental engineering problems, including water quality modeling [24, 25] and air quality modeling [26]. Among them, STM is a class of neural networks that can only use the current input information and historical information. Compared with other AI-based models, LSTM is powerful for modeling sequence data such as time series or natural language [27–29]. However, recursive neural networks such as LSTM and GRU process input sequences in parallel, so the cost of model training will increase with increasing length of input sequences. Moreover, distant historical memory will be forgotten, and they have a weak ability for temporal and spatial feature extraction [30]. Therefore, convolutional neural networks have attracted increasing attention in temporal and spatial sequence modeling. TCN [31], a model with a simple convolutional network architecture, is proposed to be applied to language modeling and music modeling and has demonstrated better performance than recurrent neural networks, especially in a long sequence. TCN has been proven to perform well in long sequence time series modeling. However, TCNs have not yet been applied in PI prediction. Therefore, a better TCN prediction architecture is explored to promote the predicted precision of PI in this study.

Waterfowl house environment data usually contain many sources of noise. To eliminate noise interference, extract essential feature information, and obtain high-quality data sets, experts have proposed many methods, such as WT, independent component analysis, and empirical mode decomposition [32–34]. WT can decompose time series with different resolutions and distinguish between noise and useful signals. It has been successfully applied to pattern recognition and noise elimination. Liu et al. proposed a hybrid wavelet analysis and least squares support vector regression with a Cauchy partial swarm optimization algorithm model for dissolved oxygen prediction [9]. Kumar et al. obtained better-quality denoised electrocardiogram signals by a denoising technique using a stationary WT compared with empirical mode decomposition, the Fourier decomposition method, and discrete WTs [35]. To effectively improve the GPR image resolution, Zhang et al. combined WT and F-K filtering [36]. Samuel et al. conducted a comparative analysis of a set of machine learning models, and the results showed that the best combination for predicting the streamflow into the Sobradinho Reservoir was the bootstrap, WT and neural network [37].

Based on the above studies, this paper proposes a new waterfowl house environmental quality assessment and prediction model combining WT and a temporal convolutional network. The WT can reduce the noise of original environmental data and obtain high-quality data, and the temporal convolutional network can extract the temporal and spatial features of processed data, output precise environmental quality assessment, and predict results.

## Materials and methods

### Study area and data source

The waterfowl culture farm in Haifeng County (23°05'N, 115°19'E) in Shanwei City, China was investigated in the present study. With an area of approximately 53.3 hm2, the farm is a multifunctional integrated aquacultural base integrating waterfowl breeding, seeding breeding and intensive aquaculture.

The environmental data were collected by the waterfowl culture internet of thing (IoT) online monitoring terminal system, and its architecture is shown in Fig 1. The system is equipped with a temperature sensor, ammonia concentration sensor, humidity sensor, etc.

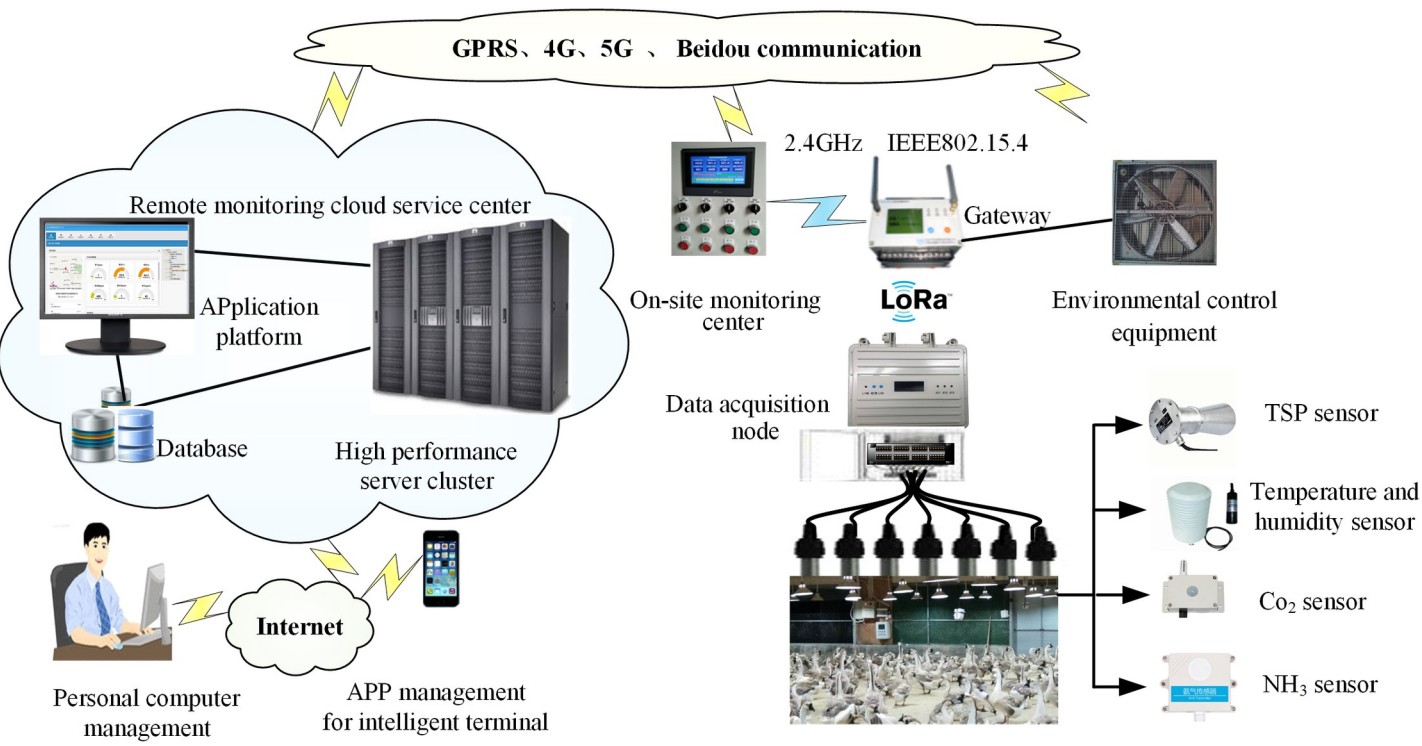

**Fig 1. Schematic diagram of the waterfowl farm monitoring system based on the internet of things.**

The environmental information is uploaded to the cloud, and users can obtain on-site information by accessing the web server through mobile phones or personal computers.

According to the importance of the environment and expert research, we selected 5 environmental factors, as shown in Table 1. Among them, ammonia is a toxic gas and the greatest concern of environmental pollution in waterfowl production, adversely affecting the ecosystem, environment, and health of birds and people. Less than 10 ppm is the ideal limit [38]. Relative humidity can impact bird health, and high relative humidity may worsen broiler geese performance; the ideal value is 70% [39]. Heat stress is a major concern in waterfowl production; high and low temperatures will reduce the growth performance and survivability of waterfowl, and temperatures less than 27˚C and more than 10˚C are ideal [40]. TSP is from the birds themselves as well as from the feed, litter, and building materials and may serve as a pathogen disseminator and bring about lung damage; less than 10 mg/m$^3$ is the ideal limit [38]. Additionally, the levels of relative humidity, ammonia, $CO_2$, temperature and TSP are known to be correlated with each other [5]. These factors are important for waterfowl production and can be monitored and optimized.

**Table 1. The selected environmental parameters for PI prediction.**

| Environmental factor | Unit | Limit | Influences |
|---|---|---|---|
| Ammonia | ppm | 10 | Ammonia is a toxic gas impairing animal performance and bird and staff health [41–42]. |
| Relative humidity | % | 70 | High relative humidity may worsen broiler geese performance [39]. |
| Temperature | ˚C | 10~27 | Influences poultry welfare and food intake, as well as increases susceptibility to disease and flock mortality rate [40]. |
| $CO_2$ | ppm | 1500 | A decrease in production and bird health can occur [41]. |
| TSP | mg/m$^3$ | 8 | A higher incidence of lung damage [38]. |

## Data sets

In this report, environmental data, including ammonia, relative humidity, temperature, $CO_2$ and TSP, were collected from July 1st to September 28th, 2019, at intervals of 20 minutes. There were 72 sets of data collected per day with a total yield of 6480 observed samples. For model generation, the first 4536 sets of data were used for model training, and the remaining 1944 sets were used as the testing data to estimate the prediction performance of the constructed model.

## Pollution index

The PI in this study was formulated within the criteria for evaluating the environmental quality of livestock and waterfowl farms by the Quality and Technology Supervision Bureau of China. The single PI can be calculated as:

$$I = \frac{C_i}{C_{si}} \tag{1}$$

where $C_i$ is the measured concentration of environmental pollutants and $C_{si}$ is the standard concentration limit of pollutants. The composite PI of waterfowl house environment quality can be described as follows:

$$P_i = \left\{ I_{max} \cdot 1/n \sum I_i \right\}^{\frac{1}{2}} \tag{2}$$

where $I_{max}$ is the maximum single PI among all pollutants, $n$ is the number of pollutants, and $I_i$ is the single PI of pollutant $i$.

## Wavelet transform

Continuous WTs for a given waterfowl environment signal $s(k)$ can be described as follows:

$$WTs(a,b) = \frac{1}{\sqrt{a}} \int_{\infty}^{-\infty} s(k) \overline{\left( \psi \left( \frac{t-b}{a} \right) \right)} dx \tag{3}$$

where $\overline{()}$ denotes the complex conjugate, $\psi(x)$ is the wavelet function, $a$ is the time scale dilation and $b$ is the time translation. By controlling the values of parameters $a$ and $b$, signal time-frequency positioning can be achieved. The WT can decompose signals into multiple resolutions. However, the symbols in realization communication are discrete data, so the continuous WT needs to be discretized. Suppose $a = a_0^j, b = kb_0 a_0^j, j, k \in Z$, when $a_0 = 2, b_0 = 1$:

$$\psi_{j,k}(n) = 2^{\frac{-j}{2}} \psi(2^{-j}n - k) \tag{4}$$

The discrete waterfowl environment signal $s(n)$ can be transformed as:

$$D(j,k) = \sum_k s(n) \overline{\psi_{j,k}}(n) \tag{5}$$

The Mallat algorithm can quickly calculate the orthogonal WT coefficients and realize signal decomposition and reconstruction [43]. The approximation coefficients $s_{j+1}(n)$ and detail

coefficients $d_{j+1}(n)$ can be recurrently related by the Mallat algorithm as follows:

$$s_{j+1}(k) = \sum_n h_0(n - 2k)s_j(n) \tag{6}$$

$$d_{j+1}(k) = \sum_n h_1(n - 2k)s_j(n) \tag{7}$$

where $h_0$ and $h_1$ are the high-pass filter and low-pass filter, respectively. The reconstruction is the inverse process of decomposition:

$$s_j(k) = \sum_n s_{j+1}(n)h_0(k - 2n) + \sum_n d_{j+1}(n)h_1(k - 2n) \tag{8}$$

Different environmental factors may contain different kinds of noise. To achieve the best effect of the denoising process and obtain quality data, we adopt four wavelet functions, Db4, Haar, Coif, and Sym10, to process each environmental dataset. The signal-noise ratio (SNR) and normalized cross correlation (NCC) were used to evaluate the denoised effect, which can be described as:

$$SNR = \frac{I_1^2}{(I_2 - I_1)^2} \tag{9}$$

$$NCC = \frac{\sum (I_1 - mean(I_1)) \cdot (I_2 - mean(I_2))}{\sqrt{\sum (I_1 - mean(I_1))^2 \times \sum (I_2 - mean(I_2))^2}} \tag{10}$$

where $I_1$ is the denoised signal and $I_2$ is the original signal; theoretically, the larger the SNR and NCC are, the better the noise reduction effect.

## Support vector machine

SVM has good generalization ability in solving nonlinear, small sample, and high-dimensional pattern recognition, and the optimal solution obtained is global, which solves the local optimal problem that cannot be avoided in other algorithms. The prediction process of the SVM includes support vector determination, kernel function selection, kernel parameter determination, and solution.

## Random forest

RF is a nonlinear ensemble model that establishes and averages a large number of random distribution decision trees for regression or classification tasks [44]. A decision tree or classification and regression tree that constructs the RF is a nonparametric model. According to the complexity of the input data, the tree grows in the process of learning. Decision nodes and leaf nodes are the main components of the decision tree. Each input sample is estimated by a test function of decision nodes and passed to different branches according to the features of the sample. After all trees are trained, each tree can predict the test sample set according to the node threshold, and the results of each tree are combined to vote to determine the final result of the entire random forest.

## Long short-term memory neural network

The LSTM neural network is a special kind of recurrent neural network. It was first proposed by Ho-chreiter and Schmidhuber [45]. Its appearance effectively solved the gradient explosion

problem of traditional recurrent neural networks. At the same time, the LSTM neural network has long-term memory to handle long-term sequence data.

## Gated recurrent unit

A GRU is a type of recurrent neural network (RNN). Similar to a long short-term memory neural network (LSTM), it is also proposed to solve long-term and gradient backpropagation problems. LSTM and GRU have similar performances, but compared with LSTM, GRU is computationally cheaper.

## Temporal convoluted network

TCN, like LSTM, is a novel neural network architecture that can be used for time series prediction. The outstanding advantage of TCNs is that they not only have much longer memory but also have higher computational efficiency than LSTM and other recurrent neural networks [46].

In general, a nature sequence modeling task is any function $f:X^{T+1} \rightarrow Y^{T+1}$ that produces the mapping:

$$y_0, y_1, ..., y_T = f(x_0, x_1, ..., x_t) \tag{11}$$

The goal of the sequence model is to fit this function $f$ to minimize the expected loss. It satisfies the causal constraint that $y_t$ depends only on $x_0, x_1, ..., x_t$ and not on future inputs $x_{t+1}$ and that the output has the same length as the input.

As shown in Fig 2, to satisfy the causal constraint, TCN uses a 1D fully convolutional network architecture [47], which is different from the traditional convoluted neural network in that the value at time $t$ only depends on the value at time $t$ and before in the previous layer. In addition, zero padding of length (kernel size—1) is added to keep subsequent layers the same length as previous layers.

One of the goals of TCNs is a long effective history size, which means an extremely deep network or very large filters. However, more convolution layers or larger filters bring about the problems of disappearing gradients, complex training, and poor fitting effects. To solve the problems above, dilated convolutions [31] were employed in the TCN. Specifically, given a sequence input $X^{n+1} = \{x_0, x_1, ..., x_n\}$ and convolution function $f : \{0, 1, ..., k-1\} \rightarrow \mathbb{R}$, the dilated convolution operation was defined as follows:

$$F(s) = \sum_{i=0}^{k-1} f(i) \cdot x_{s-i \cdot d} \tag{12}$$

where $k$ is the kernel size, $d$ is the dilated factor, and $s-i \cdot d$ accounts for the direction of the past. When $d = 1$, a dilated convolution is equal to a regular convolution. As shown in Fig 3, as the number of layers increases, the dilated factor $d$ grows, and the top layer can represent a wider range of inputs. On the other hand, choosing a larger kernel size $k$ of the filter can also effectively expand the receptive field of a ConvNet.

Residual connections proved to be an effective method for deep network training to converge quickly and reduce the risk of overfitting [48]. As shown in Fig 4, the residual block used in the TCN has two branches from input to output. The first branch contains a series convolution layer, parameter regularization, rectified linear unit, and dropout layer in order. This was a flexible architecture that allows the layer to modify parameters such as the activation function and dropout rate. The second branch ensures that the output sequence length is equal to the length of the input: if the lengths of the input sequence and output sequence are equal, the

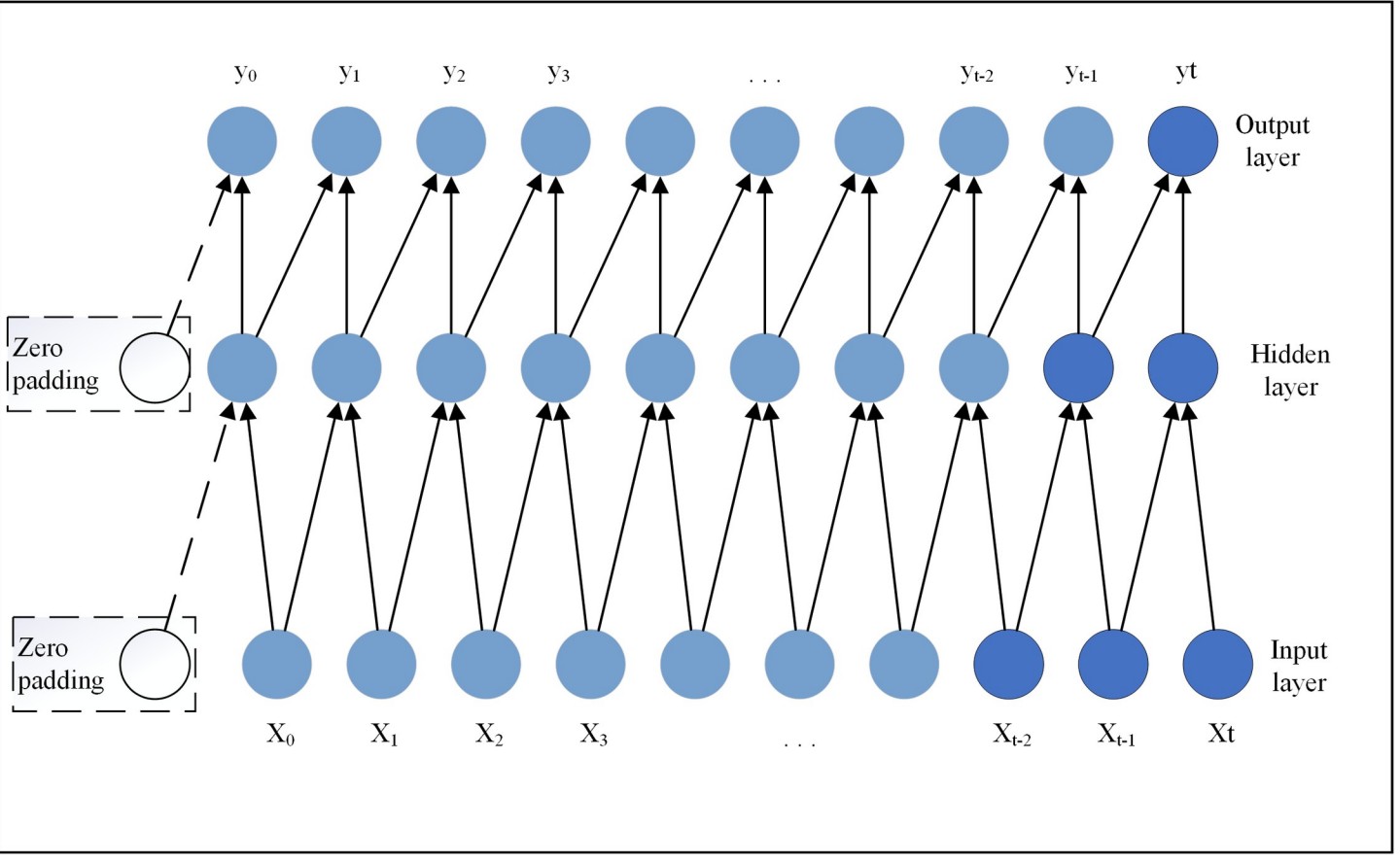

**Fig 2. Causal convolution construction.**

output layer is connected to the input layer through identity mapping; otherwise, the output layer is connected to the input layer through a 1×1 convolution.

## Performance criteria

In this paper, the mean absolute error (MAE), root mean square error (RMSE) and coefficient of determination ($R^2$) were selected to measure the prediction accuracy and operation efficiency, and MAE, RMSE and $R^2$ are defined as follows:

$$MAE = \frac{1}{N} \sum_{i=1}^{N} |y_i - y_{i'}| \tag{13}$$

$$RMSE = \sqrt{\frac{1}{N} \sum_{i=1}^{N} (y_i - y_{i'})^2} \tag{14}$$

$$R^2 = \frac{\left( \sum_{i=1}^{N} (y_i - \overline{y})(y_{i'} - \overline{y}') \right)^2}{\sum_{i=1}^{N} (y_i - \overline{y})^2 \cdot \sum_{i=1}^{N} (y_{i'} - \overline{y}')^2} \tag{15}$$

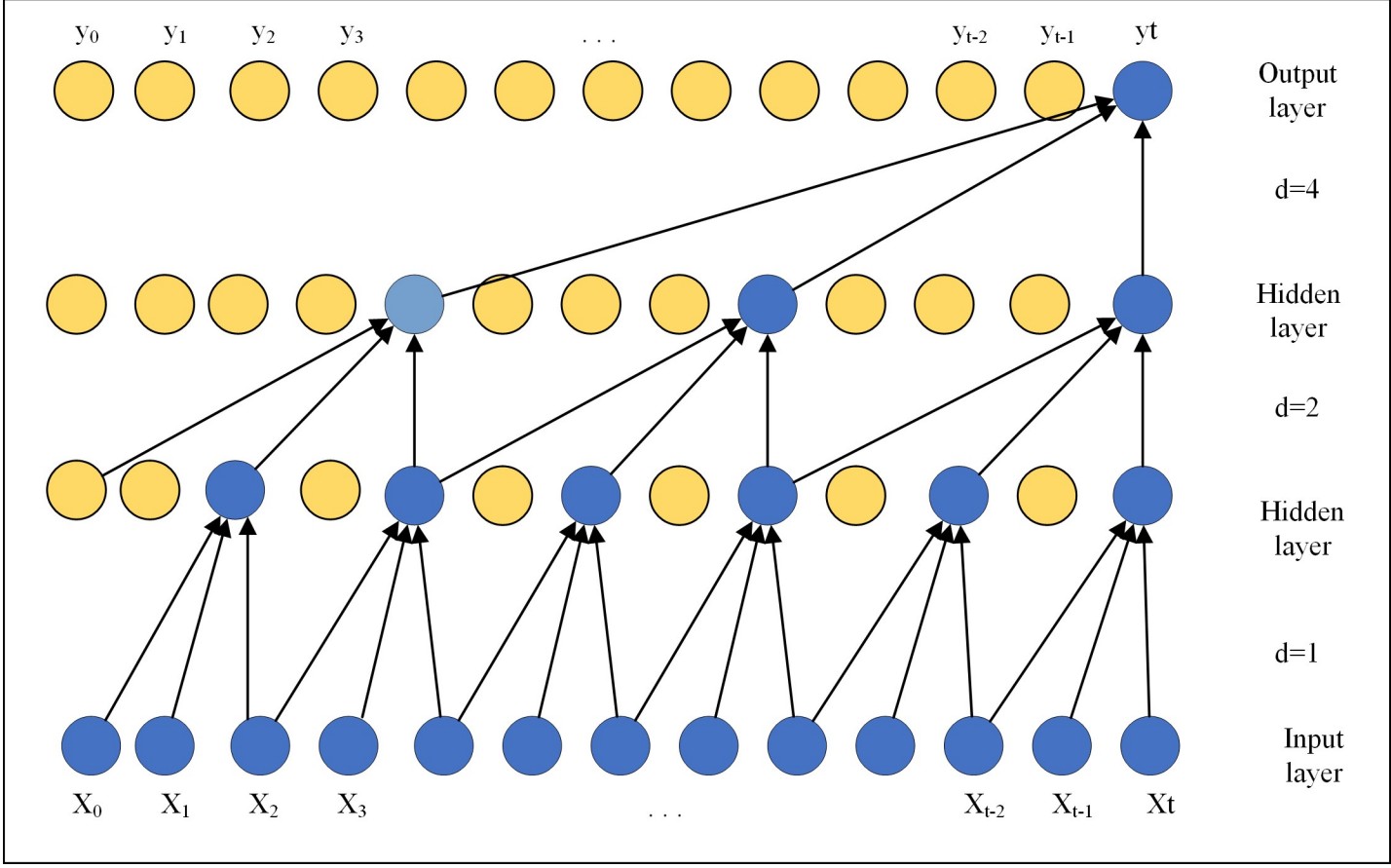

**Fig 3. Dilated convolution construction.**

where $N$ is the total number of actual points in the data, $yi$ is the observed value of period $i$, $y_i$ is the prediction value of period $i$, $y$ is the average of observed values, and $y_t$ is the average of prediction values.

## The waterfowl house environment quality assessment and prediction model

The algorithms used in this paper were implemented in Python 3.7 programming language. The equipment used in this work has an Intel Core i5-5200u processor, CPU @2.20 GHz and 8.0 GB of random access memory installed.

Both WT and TCN have the unique advantage of being able to capture data characteristics in time series. Thus, this paper uses WT-TCN to construct a model to assess and forecast the PI of waterfowl houses. The implementation process for our model is shown in Fig 5. In this study, we first reduce or eliminate the noise of environmental data by WT. Then, the denoised data and PI data were used as the input to train the TCN, and finally, the waterfowl house environment quality assessment and prediction model was obtained.

## Results and discussion

### Simulation results and discussion

As shown in Table 2, wavelet function Db4 was suitable for ammonia data (SNR = 9.4318, NCC = 0.9485), Sym10 was suitable for temperature data (SNR = 7.4635, NCC = 0.9221),

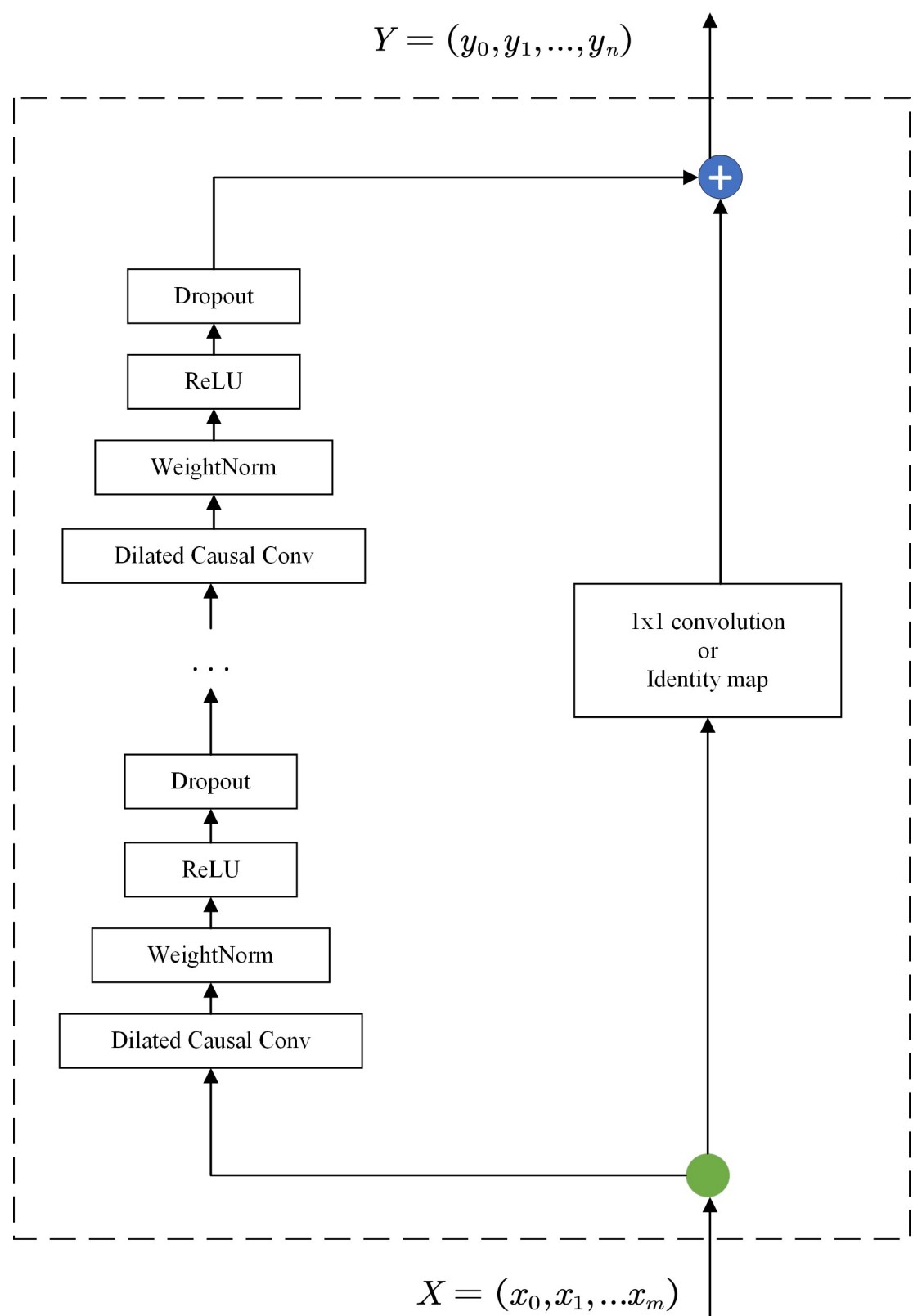

**Fig 4. An example of residual connection in a TCN.**

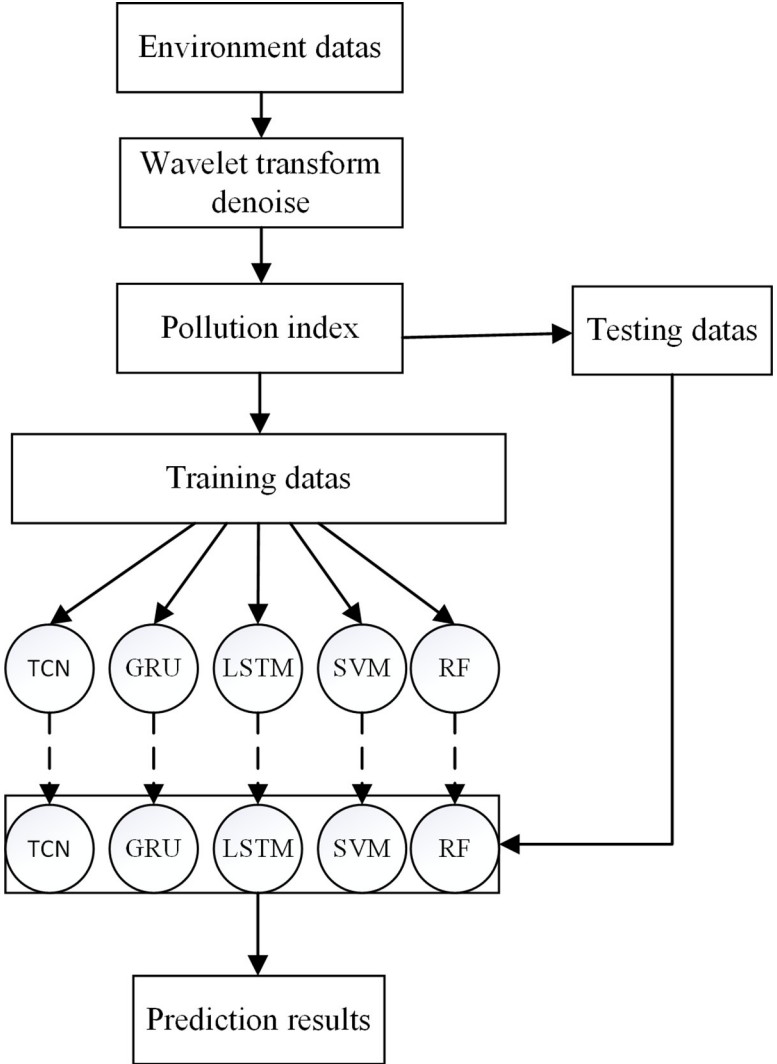

**Fig 5. Construction process of the PI prediction model.**

**Table 2. Denoised performance of five parameters.**

| Target | Performance criteria | Wavelet function | | | |
|---|---|---|---|---|---|
| | | Db4 | Haar | Coif | Sym10 |
| Ammonia | SNR/(dB) | 9.4318 | 8.8287 | 9.3769 | 9.3602 |
| | NCC | 0.9485 | 0.9432 | 0.9484 | 0.9474 |
| Temperature | SNR/(dB) | 7.2491 | 6.5372 | 7.0205 | 7.4635 |
| | NCC | 0.9192 | 0.9061 | 0.9149 | 0.9221 |
| Relative humidity | SNR/(dB) | 3.9422 | 3.6395 | 3.8920 | 4.0809 |
| | NCC | 0.8600 | 0.8559 | 0.8596 | 0.8630 |
| $CO_2$ | SNR/(dB) | 1.7064 | 1.3855 | 1.4991 | 1.7785 |
| | NCC | 0.7764 | 0.7642 | 0.7689 | 0.7793 |
| TSP | SNR/(dB) | 12.8848 | 12.6578 | 13.1443 | 13.1142 |
| | NCC | 0.9756 | 0.9745 | 0.9771 | 0.9769 |

relative humidity data (SNR = 4.0809, NCC = 0.8630), and CO2 data (SNR = 1.7785, NCC = 0.7793), and Coif was suitable for TSP data (SNR = 13.1443, NCC = 0.9771). The best denoised results of each environmental dataset are shown in Fig 6.

In this study, we tested the TCN's memory and feature extraction capability for environment data sequences of different lengths. Fig 7 shows that TCN consistently converges to approximately 0.01 MSE for all sequence lengths, whereas GRU and LSTM degenerate quickly as the sequence lengths grow. These results suggest that TCN is better at long sequence memory and feature extraction than its recurrent counterparts.

Then, TCNs were used to predict PI at different time intervals, including 12 hours, 24 hours, and 48 hours (predicting 12 hours, 24 hours and 48 hours in the future every 20 minutes). Table 3 shows that TCN has the best performance in all simulation results compared with the other models. We noted that as the prediction time interval increased, the prediction effect decreased. This can be expected because a long prediction time interval needs a much longer effective history, which is a challenge for time series models. The performance of TCN fluctuates when the prediction time interval changes from 12 hours to 24 hours, but the wave motion ranges in MAE, RMSE and $R_2$ of TCN and GRU are less than 10%. However, the wave motion range in MAE, RMSE and R2 of LSTM reached 25% when the prediction time interval changed from 12 hours to 24 hours and reached 112% when the prediction time interval changed from 24 hours to 48 hours. Moreover, the traditional SVM and RF models had poor performance in this study, which may be caused by the lack of long-time series memory ability compared with the recurrent neural network. In addition, scatter plots, Taylor plots and box plots were also used here to visualize the predictive performance of various models.

The scatter plot in Fig 8 visualized the agreement between the predicted and observed values of PI. In Fig 8, the baseline is drawn as a reference, and the perfect agreement between the observed and predicted data is described. SVM and RF models are far away from the best line. All three TCN models showed outstanding prediction performance (points close to the best line). Additionally, in the three phases, TCN is slightly far from the best line when PI reaches 3.5 or more, but GRU and LSTM perform worse. The reason for this phenomenon may be the input data of the environmental parameter $CO_2$; all were in the normal range and no data exceeded the standard value or were distributed near the border value (Fig 5(D)), leading to the trained models having low sensitivity for a high PI and prediction difficulty.

Furthermore, the models were also evaluated using Taylor plots for the three phases in Fig 9. In a Taylor figure correlation coefficient, normalized standard deviation and RMSE were drawn, and the distance between the point corresponding to the model with the best prediction performance and the "observation" point was the least. Again, Taylor plots showed that the TCN model has the best prediction performance.

Finally, Fig 10 compares the observed and predicted PI value dispersion and indicates the median ($M$) in a box plot. The performance of the three models was very similar in phase one, with the median values being very close ($M_{observed} = 1.612$, $M_{TCN} = 1.602$, $M_{GRU} = 1.648$, $M_{LSTM} = 1.719$, $M_{SVM} = 1.569$, and $M_{RF} = 1.854$), but the box shape of TCN is closer to the box shape of the observed data, which also means that all prediction results of TCN were better than the others. In both phase two and phase three, the performances of LSTM and GRU obviously worsen as the prediction interval grows, while the TCN performance does not change much. Overall, TCN substantially outperforms generic recurrent architectures such as LSTM and GRU.

In summary, TCN combines best practices such as dilations and residual connections with the causal convolutions needed for autoregressive prediction. The experimental results indicate that for all prediction time interval phases, the TCN model provides high performance for PI forecasting, especially in the long prediction time interval problem. On the other hand, the

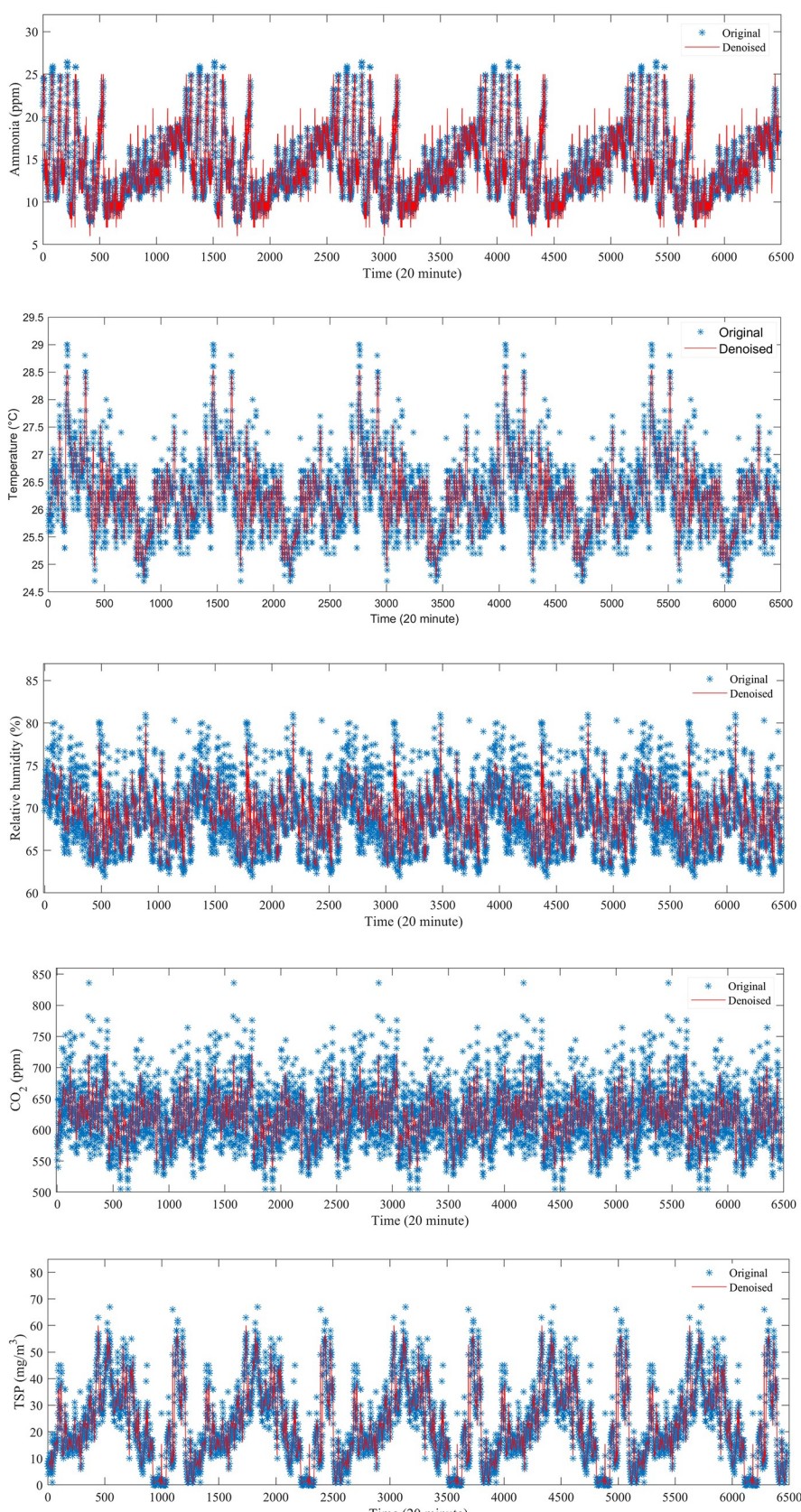

**Fig 6. Results of noise reduction of five parameters by WT.** (a) Ammonia sequence. (b) Temperature sequence. (c) Relative humidity sequence. (d) $CO_2$ sequence. (e) TSP sequence.

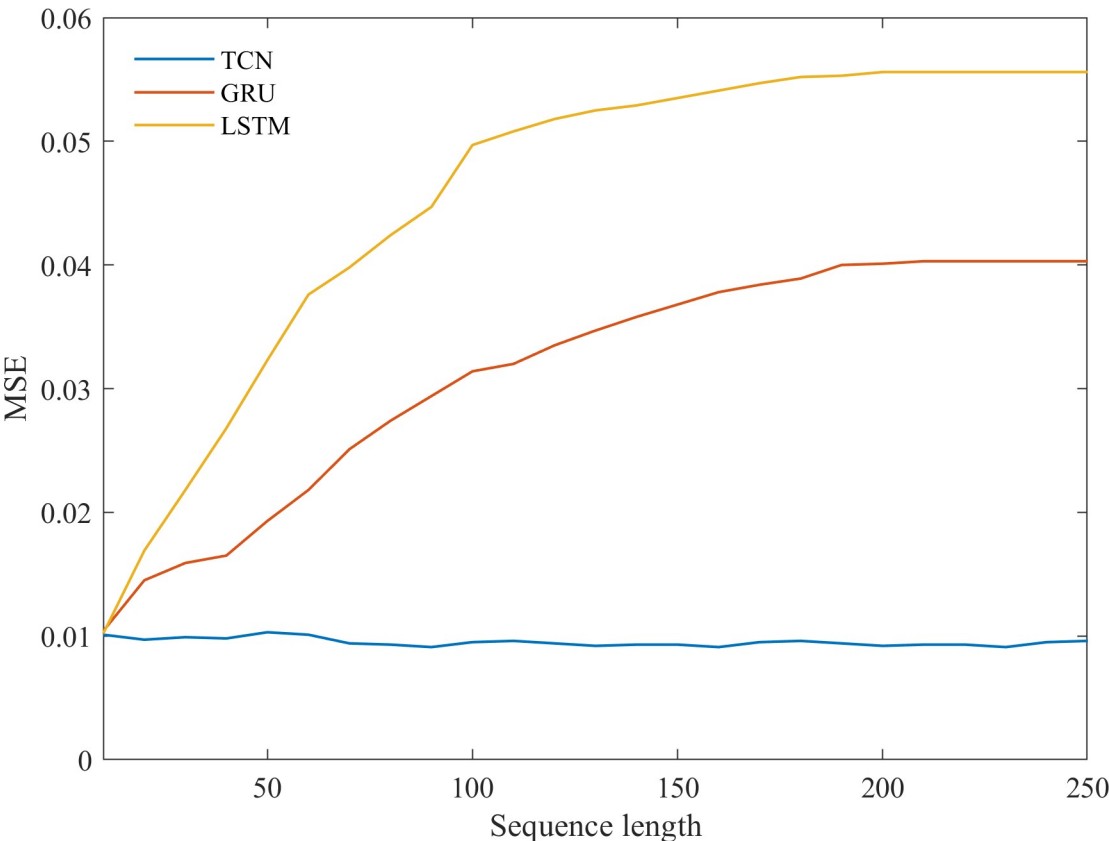

**Fig 7. Performance of three models in sequences of different lengths.**

**Table 3. Comparison of model performance.**

| | MAE | | | RMSE | | | R² | | |
|---|---|---|---|---|---|---|---|---|---|
| Hours | 12 | 24 | 48 | 12 | 24 | 48 | 12 | 48 | 48 |
| TCN | 0.0842 | 0.0859 | 0.1115 | 0.0154 | 0.0167 | 0.0273 | 0.9789 | 0.9791 | 0.9635 |
| GRU | 0.1728 | 0.1810 | 0.1922 | 0.0759 | 0.0789 | 0.0789 | 0.8937 | 0.8903 | 0.8898 |
| LSTM | 0.1892 | 0.2523 | 0.4434 | 0.0824 | 0.1388 | 0.2953 | 0.8896 | 0.8082 | 0.6078 |
| SVM | 0.5919 | 0.7245 | 0.8991 | 0.3292 | 0.3537 | 0.8181 | 0.8512 | 0.7748 | 0.6617 |
| RF | 0.2744 | 0.5794 | 0.9571 | 0.3407 | 0.3555 | 0.7513 | 0.8809 | 0.8050 | 0.6799 |

LSTM model appears to be the 'weakest' model of all three models; furthermore, it also indicates that a simple convolutional architecture is more effective across time sequence modeling tasks than recurrent architectures such as LSTM. On the other hand, a new type of temporal convoluted neural network is more competitive in the PI time series prediction of waterfowl farms than the traditional machine learning model.

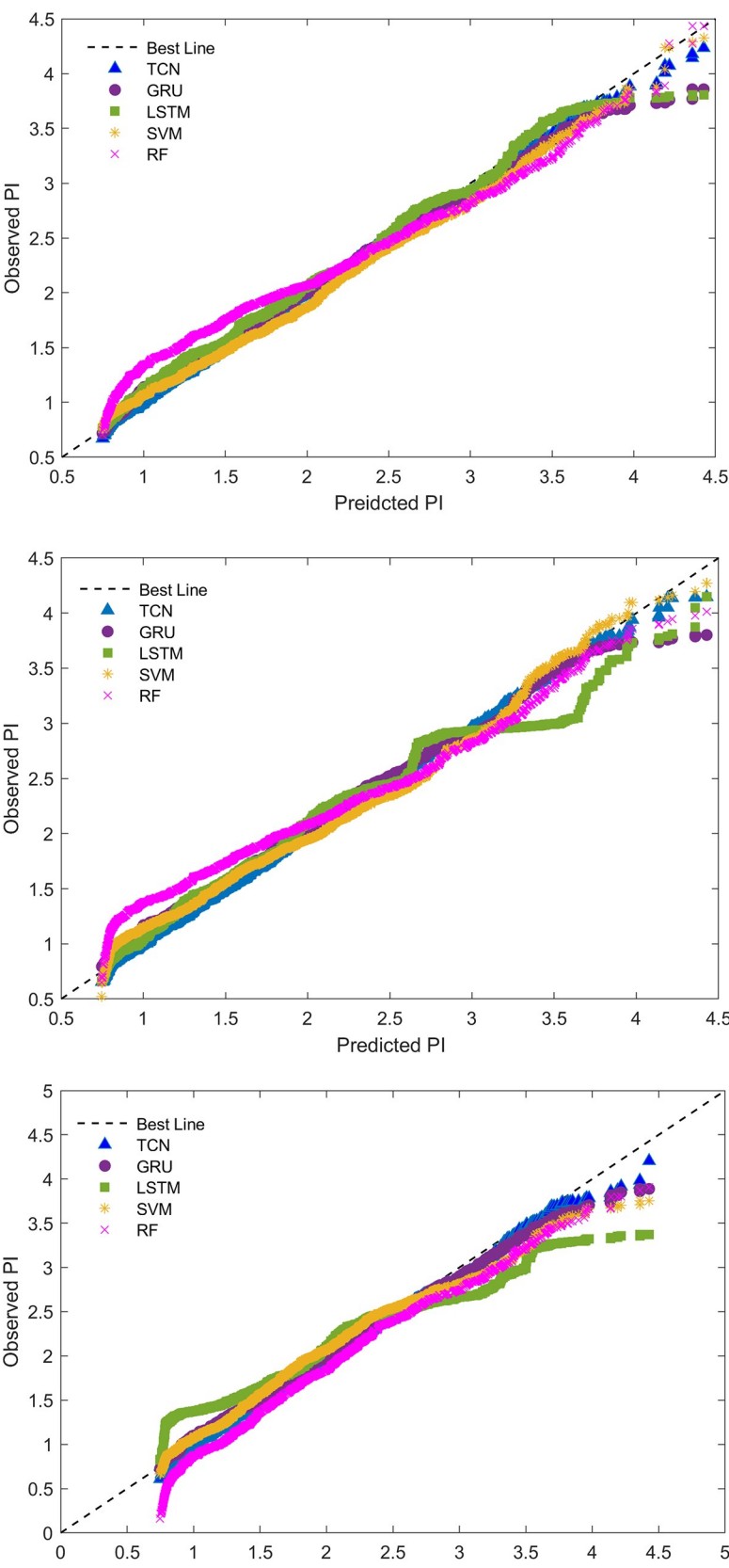

**Fig 8. Agreement between the observed and predicted PI values for the model considered in this study.** (a) Future 12-hour prediction. (b) Future 24-hour prediction. (c) Future 48-hour prediction.

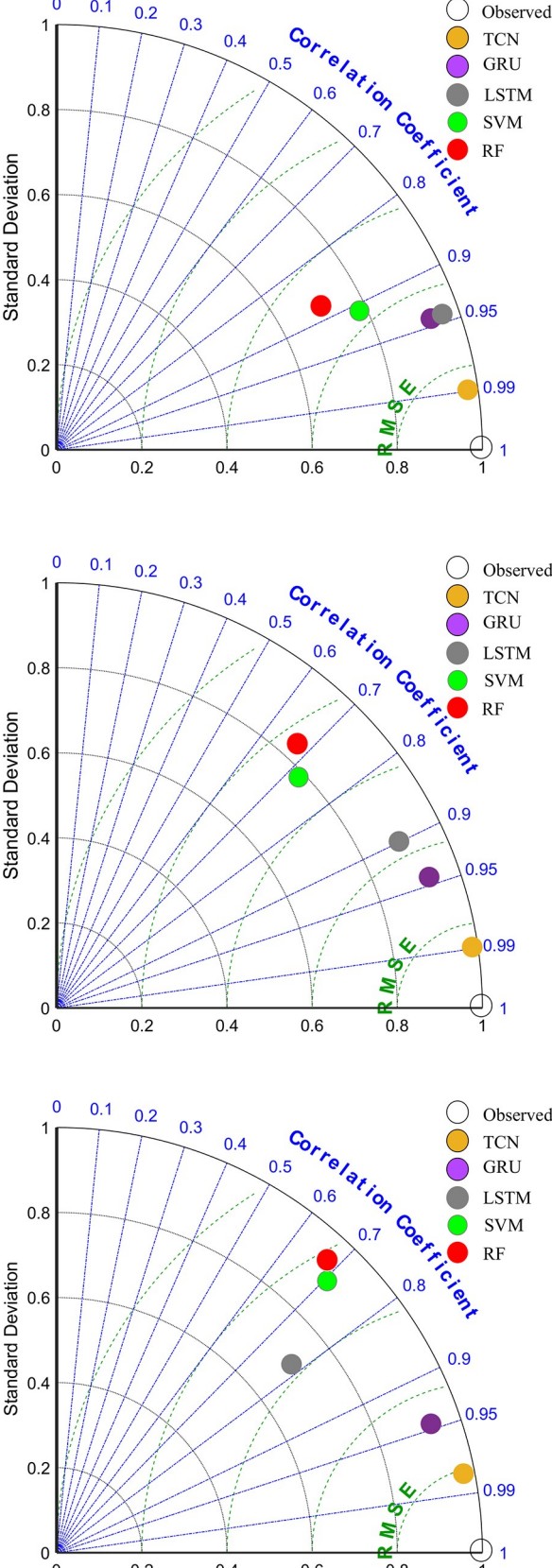

**Fig 9. Normalized Taylor diagrams.** (a) Future 12-hour prediction. (b) Future 24-hour prediction. (c) Future 48-hour prediction.

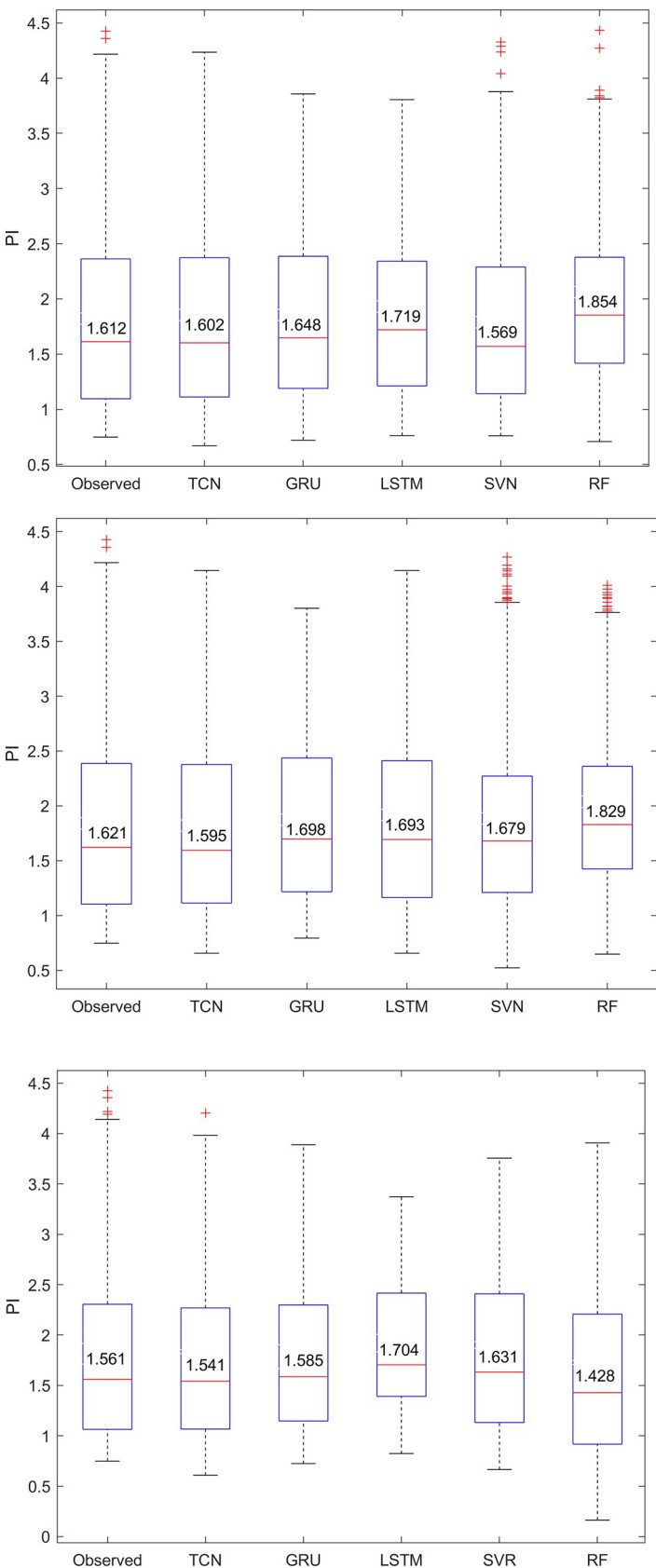

**Fig 10. Box plot of observed and predicted PI values.** (a) Future 12-hour prediction. (b) Future 24-hour prediction. (c) Future 48-hour prediction.

## Conclusions and future research

To further promote waterfowl house environment monitoring and controlling technology, reduce labor and increase the production effect, this study analyses the shortcomings of existing methods and introduces a new way to guide waterfowl house environment management by learning from other fields.

The new method investigates the application of denoised WT and the performance of three neural network models and two mechanical learning models in predicting the PI of the waterfowl house environment using environmental quality parameters at different intervals. The results indicate that the TCN model has the best performance in predicting PI. The GRU model has similar performance but lower performance when the prediction time interval changed, and the LSTM model performed the worst among the three models, although it still provided fairly accurate PI predictions.

The models presented in this paper, in particular the TCN model, could provide accurate and long-interval PI predictions of waterfowl house environments and monitor them in real time. The simulation results show that this method can be applied in waterfowl house environment prediction. The future trend of the environment can be estimated and predicted compared with traditional real-time monitoring technology, which may allow better waterfowl house environment management practices, better culture plan design, and, in general, contribute to a more sustainable waterfowl house management approach.

The environmental parameters selected in the present study may also pose a limitation because of the lack of equipment. Future work may include the use of more environmental parameters to evaluate PI and apply the model in more waterfowl breeding sites to improve the production effect and further verify the model.

## Supporting information

**S1 Dataset.**
(XLSX)

## Acknowledgments

The authors would like to thank native English experts Michele Genovese, Dr. Murtaza Hasan and AJE for revising the manuscript.

## Author Contributions

**Conceptualization:** Jiande Huang, Shuangyin Liu, Shahbaz Gul Hassan.

**Data curation:** Jiande Huang, Shuangyin Liu.

**Formal analysis:** Shuangyin Liu.

**Funding acquisition:** Shuangyin Liu.

**Project administration:** Longqin Xu.

**Resources:** Shuangyin Liu, Longqin Xu.

**Software:** Longqin Xu.

**Supervision:** Shahbaz Gul Hassan.

**Validation:** Jiande Huang.

**Visualization:** Jiande Huang.

**Writing – original draft:** Jiande Huang, Shahbaz Gul Hassan.

**Writing – review & editing:** Shuangyin Liu, Shahbaz Gul Hassan.

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
