## [Decision Letter · Decision Letter 0]

23 Feb 2021

PONE-D-21-03161

Pollution index of poultry farm assessment and prediction based on temporal convoluted network

PLOS ONE

Dear Dr. Liu,

Thank you for submitting your manuscript to PLOS ONE. After careful consideration, we feel that it has merit but does not fully meet PLOS ONE’s publication criteria as it currently stands. Therefore, we invite you to submit a revised version of the manuscript that addresses the points raised during the review process.

We look forward to receiving your revised manuscript.

Kind regards,

Chi-Hua Chen, Ph.D.

Academic Editor

PLOS ONE

3. We note you have included a table to which you do not refer in the text of your manuscript. Please ensure that you refer to Table 3 in your text; if accepted, production will need this reference to link the reader to the Table.

Reviewers' comments:

Reviewer's Responses to Questions

**Comments to the Author**

1. Is the manuscript technically sound, and do the data support the conclusions?

Reviewer #1: Yes

Reviewer #2: Partly

2. Has the statistical analysis been performed appropriately and rigorously? 

Reviewer #1: N/A

Reviewer #2: No

3. Have the authors made all data underlying the findings in their manuscript fully available?

Reviewer #1: No

Reviewer #2: No

4. Is the manuscript presented in an intelligible fashion and written in standard English?

Reviewer #1: No

Reviewer #2: No

5. Review Comments to the Author

Reviewer #1: This manuscript develops a method to apply AI to predict a pollution index for a poultry farm in China. While I think the methods have promise, the authors need to address the following major issues before it is suitable for publication.

Major:

> The English and grammar are fair to poor throughout. I have included in the minor comments below some instances in just the abstract and first couple of paragraphs but I do not have time to edit the entire manuscript. This needs to be addressed as it made it difficult to follow along.

> The motivation for the paper is lacking. Why do we need such a robust prediction method? Can the facilities actually use it to make changes? If so, does the interval required to make adjustments align with the model's ability to predict (Figure 8). For example L 58-59 is the only place you mention that it would benefit farmers, but would it? In what way? How could it be used?

> At its most basic premise, this paper is quite simple. Calculate a very simple pollution index from a few variables and see if you can predict with ML methods. It would benefit by comparing these computationally intensive methods to simpler prediction methods...does it even warrant the use of ML? What about other pollution indices? Or individual variables? Another example is you state that (L 56-58) traditional methods have bad performance. So why not show that with your data to really make the case for your work??

> L73-74: You state that TCN can better solve the modeling problem associated with PI, but you have yet to establish what those problems are. Is it just the noise which you discuss in the next paragraph? If so, why not compare the traditional methods with denoised data as well to see if the denoising was the most helpful thing or the ML approach?

> L92-95: no need to outline the paper. Delete these lines.

> Section 2.1 --you need a LOT more information on the study area. Include a map of where Shanwei is. L 99-101 describe the system but a schematic would be helpful including where the sampling is occurring. I'm also confused about what kind of animal is in the farm. A couple of times the paper mentions chicken but here it says waterfowl. So, what kind of animal is in the farm and how many are there typically? What kind of structure is the facility? Are their air ventilation systems, etc. that can help to control the parameters being measured? The data source section is missing a lot of information. I cannot tell what most of the things in Figure 1 are. It needs a narrative or more details. What kinds of sensors were there? What methods were they using to detect? What is the time frame of collection? What season did you collect data?

> Section 2.4 you need to include descriptions of all of the methods you used, not just TCN (eg LSTM, GRU)

> You have SO MANY methods in the results. All of sections 4.1, 4.2, 4.3, then lines 251-255 of section 4.4 are methods and need to be in the methods section.

> It's not clear from your methods how you split up to test the prediction of future time intervals. What time intervals were used for training? For example were your training sets 12 hour intervals trying to predict future 12, 24, and 48? It's unclear

> There is a severe lack of discussion here. What does it mean that you can predict reasonably well the future 12 hours? How does the fact that your data come from a very short time period in one year affect your results? Why do all of your predictions fail in the higher levels of pollution indices? It would be helpful to provide plots (even if in SI) of the individual data that went into the PI predictions. When the PI was high and difficult to predict, which variables were responsible, and how could this help improve future predictions? There is a lot of information hiding behind the PI calculation that would be useful to readers and for you to dive into.

> Your last line 329-330 leaves me wondering what international guidelines you are referring to and would your study have been better if you used them, so why didn't you use them and what are they?

Figures:

1 - very confusing and not enough detail. Again you need to describe the equipment used to obtain the data and the methods on board.

5 - I don't think this is quite right. Shouldn't environmental data that has been wavelet transformed have an arrow going into the pollution index, and not into the TCN? And didn't you calculate GRU and LTSM as well so where are they?

Figure 6 - units for data? what kind of range do these data have? What is the x-axis?

Minor (only for abstract and first paragraph):

> L 20-21: The first sentence is very strong, do you have anything to back up this claim? If not, reword.

> L21-22: Sounds like you invented TCN and WT, which I don't think is the case so reword. I think what you mean is your study uses/applied these methods.

> L23: the poultry to a poultry

> L23: Shanwei may not mean anything to people outside of China so suggest saying China or in Shanwei, China

> L24: is to was

> L26: Probably better to use "environmental" than "environment" when referring to the data you used throughout.

> L27: Co2 should be CO2 (with the 2 as a subscript)

> L27: I've never heard it called "total particulate suspended". Do you mean total suspended particles?

> L27: con-structing is just constructing

> L27: is to are

> L28: "the" in front of TCN

> L29: prediction

> L29: phases

> L32: generalization to general

> L32: per-formance is just performance

> In general, do not use acronyms in your abstract if you never use them later (eg. CO2, TPS, GRU, LTSM)

> L37-38: This first sentence is just odd. Suggest rewording to something like, "In response to growing global demand for food (or do you mean specifically poultry?), the Chinese poultry industry has grown to be a leader in meat and egg production".

> L38-39: The first clause adds nothing to the sentence and so your sentence doesn't make sense. Why is there such a concern that poultry need to be grown in a controlled environment? You have not made that clear.

> L39-41: "In this sense" should be "because of this desire/need/concern" if you correct the sentence before. Then you have a sort of run-on here. I think what you are trying to say is that poultry farms need to be controlled environments that can maintain a healthy ecosystem and farm productivity? or something like that.

> L41-44: reword to "Thus the main goal of controlled environments for poultry production are to maintain optimal conditions for survival and growth by establishing a habitat similar to their typical natural environment." And with this sentence as such you may not need the sentence before as it would be redundant.

Reviewer #2: I have provided below some suggestions. But, the poor quality of the language did not help me to make a comprehensive review. I highly encourage the authors to reach out to some colleagues for help and get some editing done before re-submission. The topic is interesting, but, the manuscript contains mainly some generic information about the WT which are well known. Also, there should be more emphasis on the need for this study on global scale. please make some review work on this topic for other countries of the world.

Line 201: It is not clear, what is the projection time for these analyses! Authors mentioned “forecast the pollution index of waterfowl house”. Pleas provide an appropriate time scale.

On page 14, authors mentioned “Perform wavelet transform on the original signal s(k) to obtain wavelet coefficients of various scales.” These are very basic or generic statement about the WT. Please provide most useful information only related to this study. Authors are also encouraged to see other applications of WT in different fields and cite.

It will be useful if the authors could bring up a flow chart to guide the readers about step-by-step processes of making forecasts or projection which is the aim of this study.

It will be good if the authors take some serious time for a careful proofreading. There are some errors which automated word processor grammar and corrections can detect.

Fig. 10 outcome needs to be discussed well, being the key conclusion.

Once the pollutants are out in the atmosphere, there is also important consequences for the society. There are plenty of literature on this topic as well. But, I referred to one of them below.

A novel approach for the characterisation of transport and optical properties of aerosol particles near sources–Part I and Part II

6. PLOS authors have the option to publish the peer review history of their article (what does this mean?). If published, this will include your full peer review and any attached files.

Reviewer #1: No

Reviewer #2: No

---

## [Author Response · Author response to Decision Letter 0]

31 Mar 2021

Response to Journal:

Answer: We have modified the format following your suggestion. Please see the highlighted version of the manuscripts.

Answer:Following your suggesstions we have updated ORCID.

3. We note you have included a table to which you do not refer in the text of your manuscript. Please ensure that you refer to Table 3 in your text; if accepted, production will need this reference to link the reader to the Table.

Answer: We have modified the changes following your suggesstion, which are:

Table.3 shown that TCN has the best performance in all simulation results compared with other

Reviewer #1: This manuscript develops a method to apply AI to predict a pollution index for a poultry farm in China. While I think the methods have promise, the authors need to address the following major issues before it is suitable for publication.

Major:

> The English and grammar are fair to poor throughout. I have included in the minor comments below some instances in just the abstract and first couple of paragraphs but I do not have time to edit the entire manuscript. This needs to be addressed as it made it difficult to follow along.

Answer: Thank you very much for you prestigious comments we have modified the manuscript and made changes following your suggestions which are given below.

Comment: The motivation for the paper is lacking. Why do we need such a robust prediction method? Can the facilities actually use it to make changes? If so, does the interval required to make adjustments align with the model's ability to predict (Figure 8). For example, L58-59 is the only place you mention that it would benefit farmers, but would it? In what way? How could it be used?

Answer: Provide a good growing environment for the goose to ensure the normal growth and development of the goose. For example, by predicting future changes in PI the farmer can make arrangements for the work time or power of environmental control equipment such as fans, exhaust fans, and/or nebulizers in advance.

Comment: At its most basic premise, this paper is quite simple. Calculate a very simple pollution index from a few variables and see if you can predict with ML methods. It would benefit by comparing these computationally intensive methods to simpler prediction methods...does it even warrant the use

of ML? What about other pollution indices? Or individual variables? Another example is you state that (L56-58) traditional methods have bad performance. So why not show that with your data to really make the case for your work?

Answer: Following your suggestion we make comparison with tradition method and the results are given below:

Table.3 Comparison of model performance.

 MAE RMSE R2

Hours 12 24 48 12 24 48 12 48 48

TCN 0.0842 0.0859 0.1115 0.0154 0.0167 0.0273 0.9789 0.9791 0.9635

GRU 0.1728 0.1810 0.1922 0.0759 0.0789 0.0789 0.8937 0.8903 0.8898

LSTM 0.1892 0.2523 0.4434 0.0824 0.1388 0.2953 0.8896 0.8082 0.6078

SVM 0.5919 0.7245 0.8991 0.3292 0.3537 0.8181 0.8512 0.7748 0.6617

RF 0.2744 0.5794 0.9571 0.3407 0.3555 0.7513 0.8809 0.8050 0.6799

Comm

---

## [Decision Letter · Decision Letter 1]

26 Apr 2021

PONE-D-21-03161R1

Pollution index of waterfowl farm assessment and prediction based on temporal convoluted network

PLOS ONE

Dear Dr. Liu,

Thank you for submitting your manuscript to PLOS ONE. After careful consideration, we feel that it has merit but does not fully meet PLOS ONE’s publication criteria as it currently stands. Therefore, we invite you to submit a revised version of the manuscript that addresses the points raised during the review process.

We look forward to receiving your revised manuscript.

Kind regards,

Chi-Hua Chen, Ph.D.

Academic Editor

PLOS ONE

Reviewers' comments:

Reviewer's Responses to Questions

**Comments to the Author**

1. If the authors have adequately addressed your comments raised in a previous round of review and you feel that this manuscript is now acceptable for publication, you may indicate that here to bypass the “Comments to the Author” section, enter your conflict of interest statement in the “Confidential to Editor” section, and submit your "Accept" recommendation.

Reviewer #1: (No Response)

Reviewer #2: All comments have been addressed

2. Is the manuscript technically sound, and do the data support the conclusions?

Reviewer #1: Partly

Reviewer #2: Yes

3. Has the statistical analysis been performed appropriately and rigorously? 

Reviewer #1: Yes

Reviewer #2: Yes

4. Have the authors made all data underlying the findings in their manuscript fully available?

Reviewer #1: No

Reviewer #2: Yes

5. Is the manuscript presented in an intelligible fashion and written in standard English?

Reviewer #1: No

Reviewer #2: Yes

6. Review Comments to the Author

Reviewer #1: While I commend the authors for working diligently on the suggested comments by both reviewers, the manuscript is still not in a publishable state. I still maintain that the work is interesting and prudent, but the authors have not fully addressed some of the major comments brought up previously. First being the English is still not at a level of an academic paper and really needs major edits which I cannot put forth the time to do. Second, the paper is still very methods focus despite the fact that both reviewers pointed out that there needs to be more on the "why" do the study. What's the motivation? How can it help farmers. The authors added a few lines here and there but it is still not compelling. Other things that have come up with this new version include:

1. While the first table highlights the data chosen for the PI. The text does not go into much detail about why these were chosen, and what are the acceptable limits. I still do not see a mention of other similar studies on waterfowl or poultry in the introduction.

2. There is now way too much detail in the methods. I'm not sure you need every detail and plot for each AI technique in your paper unless this is the first time they are introduced. You should provide a brief summary of each and then if you want to include this detail move to SI.

3. Conclusions and future research again focus way too much on methods and not on the "why".

Reviewer #2: All comments were addresses and could be accepted as it is. I do not have additional comments at this stage.

7. PLOS authors have the option to publish the peer review history of their article (what does this mean?). If published, this will include your full peer review and any attached files.

Reviewer #1: No

Reviewer #2: No

---

## [Author Response · Author response to Decision Letter 1]

6 May 2021

Response to Journal:

Reviewer #1: While I commend the authors for working diligently on the suggested comments by both reviewers, the manuscript is still not in a publishable state. I still maintain that the work is interesting and prudent, but the authors have not fully addressed some of the major comments brought up previously. First being the English is still not at a level of an academic paper and really needs major edits which I cannot put forth the time to do. 

Answer: Thank you very much for your prestigious comments. This paper was edited for proper English language, grammar, punctuation, spelling, and overall style by one or more of the highly qualified native English speaking editors at AJE.

Second, the paper is still very methods focus despite the fact that both reviewers pointed out that there needs to be more on the "why" do the study. What's the motivation? How can it help farmers. The authors added a few lines here and there but it is still not compelling. 

Answer: We have added more references to this study. It includes why we need to monitor and control the poultry house environment, what methods have been used to do that thing by others, how can they help the farmer and why we need the new method.

Labor shortages and increasing biosecurity practices will make it more difficult for producers to monitor and manage the production, health, and welfare status of all of their birds. Employing modern poultry management technology is necessary to increase production [6]. An example of how modern management technology can be used to monitor and control the poultry house environment is exemplified by humidity regulation via ventilation rate changes mediated by relative humidity sensors, as relative humidity is one of the more important environmental aspects of a poultry house [7]. In addition, more advanced systems are being researched. Bustamante et al. used a multisensor system to effectively assess barn ventilation system function by tracking temperature, air velocity and differential pressure in broiler houses [8]. Hanif et al. proposed an internet of things technology-based protection and monitoring of the environment of a poultry house to monitor the environment-related parameters such as air temperature, air humidity, CO2 level of concentration and ammonia concentration., which has been implemented successfully, leading to a safe environment and profit for the poultry industry [9]. The techniques mentioned above are both solutions for real-time environmental monitoring and control; however, relying on hardware monitoring in real time cannot capture the trend of environmental changes [10], and it is easy to miss the best time for adjustment, which leads to waterfowl health damage and property loss.

Cultivation environment forecasting has been studied for many years and has made some achievements in aquaculture and livestock breeding [11–14]. The technique estimates or predicts the future changes in target variables that cannot be obtained directly. For example, Jackman et al. generated a prediction model by using sensor inputs of relative humidity, carbon dioxide, temperature, and ammonia for environmental parameter and crop yield prediction [15]. In waterfowl production, a system such as this would allow for actions to be taken sooner by farmers if the environment is projected to be bad. However, few studies have applied prediction in waterfowl breeding. Therefore, it is necessary to apply environmental prediction technology to waterfowl production to fill this gap.

1. While the first table highlights the data chosen for the PI. The text does not go into much detail about why these were chosen, and what are the acceptable limits. I still do not see a mention of other similar studies on waterfowl or poultry in the introduction.

Answer: We have added more references to detail describe why data were chosen, and what are the acceptable limits.

Labor shortages and increasing biosecurity practices will make it more difficult for producers to monitor and manage the production, health, and welfare status of all of their birds. Employing modern poultry management technology is necessary to increase production [6]. An example of how modern management technology can be used to monitor and control the poultry house environment is exemplified by humidity regulation via ventilation rate changes mediated by relative humidity sensors, as relative humidity is one of the more important environmental aspects of a poultry house [7]. In addition, more advanced systems are being researched. Bustamante et al. used a multisensor system to effectively assess barn ventilation system function by tracking temperature, air velocity and differential pressure in broiler houses [8]. Hanif et al. proposed an internet of things technology-based protection and monitoring of the environment of a poultry house to monitor the environment-related parameters such as air temperature, air humidity, CO2 level of concentration and ammonia concentration., which has been implemented successfully, leading to a safe environment and profit for the poultry industry [9]. The techniques mentioned above are both solutions for real-time environmental monitoring and control; however, relying on hardware monitoring in real time cannot capture the trend of environmental changes [10], and it is easy to miss the best time for adjustment, which leads to waterfowl health damage and property loss.

Cultivation environment forecasting has been studied for many years and has made some achievements in aquaculture and livestock breeding [11–14]. The technique estimates or predicts the future changes in target variables that cannot be obtained directly. For example, Jackman et al. generated a prediction model by using sensor inputs of relative humidity, carbon dioxide, temperature, and ammonia for environmental parameter and crop yield prediction [15]. In waterfowl production, a system such as this would allow for actions to be taken sooner by farmers if the environment is projected to be bad. However, few studies have applied prediction in waterfowl breeding. Therefore, it is necessary to apply environmental prediction technology to waterfowl production to fill this gap.

According to the importance of the environment and expert research, we selected 5 environmental factors, as shown in Table 1. Among them, ammonia is a toxic gas and the greatest concern of environmental pollution in waterfowl production, adversely affecting the ecosystem, environment, and health of birds and people. Less than 10 ppm is the ideal limit [42]. Relative humidity can impact bird health, and high relative humidity may worsen broiler geese performance; the ideal value is 70% [43]. Heat stress is a major concern in waterfowl production; high and low temperatures will reduce the growth performance and survivability of waterfowl, and temperatures less than 27°C and more than 10°C are ideal [44]. TSP is from the birds themselves as well as from the feed, litter, and building materials and may serve as a pathogen disseminator and bring about lung damage; less than 10 mg/m3 is the ideal limit [42]. Additionally, the levels of relative humidity, ammonia, carbon dioxide, temperature and TSP are known to be correlated with each other [6]. These factors are important for waterfowl production and can be monitored and optimized.

2. There is now way too much detail in the methods. I'm not sure you need every detail and plot for each AI technique in your paper unless this is the first time they are introduced. You should provide a brief summary of each and then if you want to include this detail move to SI.

Answer: We have simplify introduction of some methods.

Wavelet transform

Continuous wavelet transforms for a given waterfowl environment signal s(k) can be described as follows:

 (3)

where denotes the complex conjugate, is the wavelet function, a is the time scale dilation and b is the time translation. By controlling the values of parameters a and b, signal time-frequency positioning can be achieved. The wavelet transform can decompose signals into multiple resolutions. However, the symbols in realization communication are discrete data, so the continuous wavelet transform needs to be discretized. Suppose , when :

 (4)

The discrete waterfowl environment signal s(n) can be transformed as:

 (5)

The Mallat algorithm can quickly calculate the orthogonal wavelet transform coefficients and realize signal decomposition and reconstruction [47]. The approximation coefficients and detail coefficients can be recurrently related by the Mallat algorithm as follows:

 (6)

 (7)

where h0 and h1 are the high-pass filter and low-pass filter, respectively. The reconstruction is the inverse process of decomposition:

 (8)

The wavelet transform can decompose noise signals into different signal channels at various levels of resolution. In general, the useful signals are distributed in the high-level resolution channel, and the noise signal is likely to be distributed in the lower-level channel. Therefore, the noise of the original data can be reduced as much as possible based on a reasonable threshold. Therefore, the wavelet denoised algorithm can be described as follows:

(1) Perform wavelet transform on the original signal s(k) to obtain wavelet coefficients of various scales.

(2) Determine the threshold and perform threshold denoising on different wavelet coefficients to obtain estimated wavelet coefficients.

(3) Conduct wavelet reconstruction by equation 11 to obtain the denoise signal .

Support vector machine

SVM has good generalization ability in solving nonlinear, small sample, and high-dimensional pattern recognition, and the optimal solution obtained is global, which solves the local optimal problem that cannot be avoided in other algorithms. The prediction process of the SVM includes support vector determination, kernel function selection, kernel parameter determination, and solution. The mathematical expression as follow:

 (9)

Where SVS is the number of support vectors, αi is the Lagrangian coefficient of each training sample, yi (-1 or 1) is the vector label, K (xi, x) is the selected kernel function, and b0 is the bias. The kernel function replaces the linear quantity in the traditional linear equation and maps the data to high-dimensional space processing. After selected the kernel function, the samples are trained to establish the SVM model and the prediction result Nt is finally obtained.

Long short-term memory neural network

The LSTM neural network is a special kind of recurrent neural network. It was first proposed by Ho-chreiter and Schmidhuber [49]. Its appearance effectively solved the gradient explosion problem of traditional recurrent neural networks. At the same time, the LSTM neural network has long-term memory to handle long-term sequence data. The LSTM cell structure consists of forget gate, input gate, output gate and cell state. LSTM structure is shown in Fig 2 At the time t, there are three input parameters of the LSTM network: input value xt, at time t, output value ht-1 and cell state Ct-1 at t-1. There are two output parameters of the LSTM network: output value ht and cell state Ct at time t. Through the activation function σ LSTM realizes the control of the three gates so as to realize the retention and forgetting of historical information.

Fig.2. The structure of LSTM

Gated recurrent unit

A gated recurrent unit (GRU) is a type of recurrent neural network (RNN). Similar to a long short-term memory neural network (LSTM), it is also proposed to solve long-term and gradient backpropagation problems. LSTM and GRU have similar performances, but compared with LSTM, GRU is computationally cheaper. The structure of GRU is shown in Fig 3 GRU can be descried as follows:

Firstly, two output gateds, r (reset gated) and z (update gated), were obtained by the input X(t) at t time and the output h(t − 1) at t − 1 time.

 (10)

 (11)

Where CJ is a sigmoid function. Parameter h' selective re- members X(t). Trough the reset gated r we can obtain h’ as follows:

 (12)

Finally, the output y(t) and hidden state h(t) at t time can be calculated as follows:

 (13)

 (14)

Where [ ] denoted the connection of two vectors and * denoted the product of matrices.

Fig.3. The structure of GRU

3. Conclusions and future research again focus way too much on methods and not on the "why".

Answer: We have modified the conclusions.

To further promote waterfowl house environment monitoring and controlling technology, reduce labor and increase the production effect, this study analyses the shortcomings of existing methods and introduces a new way to guide waterfowl house environment management by learning from other fields.

The new method investigates the application of denoised WT and the performance of three neural network models and two mechanical learning models in predicting the pollution index of the waterfowl house environment using environmental quality parameters at different intervals. The results indicate that the TCN model has the best performance in predicting PI. The GRU model has similar performance but lower performance when the prediction time interval changed, and the LSTM model performed the worst among the three models, although it still provided fairly accurate PI predictions.

The models presented in this paper, in particular the TCN model, could provide accurate and long-interval PI predictions of waterfowl house environments and monitor them in real time. The simulation results show that this method can be applied in waterfowl house environment prediction. The future trend of the environment can be estimated and predicted compared with traditional real-time monitoring technology, which can allow better waterfowl house environment management practices, better culture plan design, and, in general, contribute to a more sustainable waterfowl house management approach.

The environmental parameters selected in the present study may also pose a limitation because of the lack of equipment. Future work may include the use of more environmental parameters to evaluate PI and apply the model in more waterfowl breeding sites to improve the production effect and further verify the model.

---

## [Decision Letter · Decision Letter 2]

25 May 2021

PONE-D-21-03161R2

Pollution index of waterfowl farm assessment and prediction based on temporal convoluted network

PLOS ONE

Dear Dr. Liu,

Thank you for submitting your manuscript to PLOS ONE. After careful consideration, we feel that it has merit but does not fully meet PLOS ONE’s publication criteria as it currently stands. Therefore, we invite you to submit a revised version of the manuscript that addresses the points raised during the review process.

We look forward to receiving your revised manuscript.

Kind regards,

Chi-Hua Chen, Ph.D.

Academic Editor

PLOS ONE

Journal Requirements:

Reviewers' comments:

Reviewer's Responses to Questions

**Comments to the Author**

1. If the authors have adequately addressed your comments raised in a previous round of review and you feel that this manuscript is now acceptable for publication, you may indicate that here to bypass the “Comments to the Author” section, enter your conflict of interest statement in the “Confidential to Editor” section, and submit your "Accept" recommendation.

Reviewer #1: All comments have been addressed

Reviewer #2: All comments have been addressed

2. Is the manuscript technically sound, and do the data support the conclusions?

Reviewer #1: Yes

Reviewer #2: Yes

3. Has the statistical analysis been performed appropriately and rigorously? 

Reviewer #1: Yes

Reviewer #2: Yes

4. Have the authors made all data underlying the findings in their manuscript fully available?

Reviewer #1: Yes

Reviewer #2: Yes

5. Is the manuscript presented in an intelligible fashion and written in standard English?

Reviewer #1: Yes

Reviewer #2: Yes

6. Review Comments to the Author

Reviewer #1: The authors have done a much better job addressing remaining concerns, especially with language. I made suggested grammatical changes in the attached document. I have some other minor things to address below, but then should be suitable for publication.

1. The order of the methods and results seems off. For example there is a wavelet transform section and then further section on wavelet transform starting on 270. Then you also have results of denoising in methods not results. I think rearranging for a better flow is needed.

2. Acronyms need to be fixed. You should define acronyms the first time only and then use the acronym after. You have many in the manuscript so this needs to be addressed.

3. Figures 6a-e are still hard to read. Maybe try removing the line on the original data so that the observed points can be seen overlain on denoised data?

Reviewer #2: Can be accepted as it is now. I do not have further comments at this stage. Authors have addressed all the comments.

7. PLOS authors have the option to publish the peer review history of their article (what does this mean?). If published, this will include your full peer review and any attached files.

Reviewer #1: No

Reviewer #2: No

---

## [Author Response · Author response to Decision Letter 2]

29 May 2021

Dear editor,

Thank you very much for your letter and advice. We have revised the paper, and would like to re-submit it for your consideration. We have addressed the comments raised by the reviewers, and the amendments are highlighted in the revised manuscript. We hope that the revision is acceptable, and I look forward to hearing from you soon.

Yours sincerely,

Shuangyin Liu 

Dean and Professor 

School of Information science and Technology

Zhongkai University of Agriculture and Engineering

Guangzhou, 510225, P.R. China; 

E-mail: shuangyinliu@zhku.edu.cn

Fax: +86-020-8900-3114;

Tel: +86-13822211958;

---

## [Decision Letter · Decision Letter 3]

22 Jun 2021

Pollution index of waterfowl farm assessment and prediction based on temporal convoluted network

PONE-D-21-03161R3

Dear Dr. Liu,

We’re pleased to inform you that your manuscript has been judged scientifically suitable for publication and will be formally accepted for publication once it meets all outstanding technical requirements.

Kind regards,

Chi-Hua Chen, Ph.D.

Academic Editor

PLOS ONE

Additional Editor Comments (optional):

Reviewers' comments:

Reviewer's Responses to Questions

**Comments to the Author**

1. If the authors have adequately addressed your comments raised in a previous round of review and you feel that this manuscript is now acceptable for publication, you may indicate that here to bypass the “Comments to the Author” section, enter your conflict of interest statement in the “Confidential to Editor” section, and submit your "Accept" recommendation.

Reviewer #1: All comments have been addressed

2. Is the manuscript technically sound, and do the data support the conclusions?

Reviewer #1: Yes

3. Has the statistical analysis been performed appropriately and rigorously? 

Reviewer #1: Yes

4. Have the authors made all data underlying the findings in their manuscript fully available?

Reviewer #1: Yes

5. Is the manuscript presented in an intelligible fashion and written in standard English?

Reviewer #1: Yes

6. Review Comments to the Author

Reviewer #1: (No Response)

7. PLOS authors have the option to publish the peer review history of their article (what does this mean?). If published, this will include your full peer review and any attached files.

Reviewer #1: No

---

## [Editor Report · Acceptance letter]

16 Jul 2021

PONE-D-21-03161R3 

Pollution index of waterfowl farm assessment and prediction based on temporal convoluted network 

Dear Dr. Liu:

I'm pleased to inform you that your manuscript has been deemed suitable for publication in PLOS ONE. Congratulations! Your manuscript is now with our production department. 

Kind regards, 

on behalf of

Professor Chi-Hua Chen 

Academic Editor

PLOS ONE